# Biodegradable Polyester Synthesis in Renewed Aqueous Polycondensation Media: The Core of the New Greener Polymer-5B Technology

Ana C. D. Pfluck [1], Dragana P.C. de Barros [2] and Luis P. Fonseca [1,3,*]

1   Institute for Bioengineering and Biosciences, Instituto Superior Técnico, Universidade de Lisboa, Avenida Rovisco Pais, 1049-001 Lisboa, Portugal; acdpfluck@gmail.com
2   Instituto de Tecnologia Química e Biológica António Xavier (ITQB), Universidade Nova de Lisboa, Avenida da Republica, 2780-157 Oeiras, Portugal; dragana@itqb.unl.pt
3   Department of Bioengineering, Institute for Bioengineering and Biosciences, Instituto Superior Técnico, Universidade de Lisboa, Avenida Rovisco Pais, 1049-001 Lisboa, Portugal
*   Correspondence: luis.fonseca@tecnico.ulisboa.pt; Tel.: +351-218-419-139

**Abstract:** An innovative enzymatic polycondensation of dicarboxylic acids and dialcohols in aqueous polymerization media using free and immobilized lipases was developed. Various parameters (type of lipases, temperature, pH, stirring type and rate, and monomer carbon chain length) of the polycondensation in an oil-in-water ($o/w$) miniemulsion (>80% in water) were evaluated. The best results for polycondensation were achieved with an equimolar monomer concentration (0.5 M) of octanedioic acid and 1,8-octanediol in the miniemulsion and water, both at initial pH 5.0 with immobilized *Pseudozyma antarctica* lipase B (PBLI). The synthesized poly(octamethylene suberate) (POS) in the miniemulsion is characterized by a molecular weight of 7800 g mol$^{-1}$ and a conversion of 98% at 45 °C after 48 h of polycondensation in batch operation mode. A comparative study of polycondensation using different operation modes (batch and fed-batch), stirring type, and biocatalyst reutilization in the miniemulsion, water, and an organic solvent (cyclohexane:tetrahydrofuran 5:1 $v/v$) was performed. Regarding the polymer molecular weight and conversion (%), batch operation mode was more appropriate for the synthesis of POS in the miniemulsion and water, and fed-batch operation mode showed better results for polycondensation in the organic solvent. The miniemulsion and water used as polymerization media showed promising potential for enzymatic polycondensation since they presented no enzyme inhibition for high monomer concentrations and excellent POS synthesis reproducibility. The PBLI biocatalyst presented high reutilization capability over seven cycles (conversion > 90%) and high stability equivalent to 72 h at 60 °C on polycondensation in the miniemulsion and water. The benefits of polycondensation in aqueous media using an o/w miniemulsion or water are the origin of the new concept strategy of the green process with a green product that constitutes the core of the new greener polymer-5B technology.

**Keywords:** enzymatic polycondensation; aqueous media; biocatalyst reutilization; stability

## 1. Introduction

Polymers have considerable importance in modern society, but they are not often considered green materials. Political and ecological regulations encourage minimizing the environmental impact by producing polymers with biodegradability characteristics. For this reason, new biodegradable polymers from renewable sources are required urgently. These greener plastics are based on polyesters, polyamides, and, sometimes, polyurethanes, which can be completely hydrolyzed to the corresponding monomers that can be reused or degraded by microorganisms or isolated enzymes [1,2].

Aliphatic polyesters with different families of poly(alkylene dicarboxylate)s are biodegradable polymers that can answer to those demands, considering that they can

be obtained using monomers from renewable resources [3–5], given that poly(alkylene dicarboxylate)s have favorable biodegradable and bio-absorbable properties and their physical and chemical characteristics can be manipulated over a broad range and have the potential for industrial and biomedical applications [6,7]. Drug delivery systems, regenerative implantation surgery, and therapeutic cell and tissue cultures are the focus of various studies for biomedical applications [8,9].

Polymer synthesis via organic chemistry has been the most used production strategy until now but with a strong negative environmental impact [10]. In general, aliphatic polyesters are produced via organic chemistry by polycondensation reactions at high temperatures and low pressure and using Lewis acids or surfactant-combined catalysts [11–13]. Microwave-assisted polycondensations in an organic solvent and solvent-free media using chemical catalysts were also reported [14]. The resulting polyesters are characterized by a lack of specificity and low solubility in biological fluids for biomedical applications and raised toxicity concerns.

To overcome these adverse effects, biocatalysis, using free or immobilized enzymes, is gaining attention as an alternative process for polyester synthesis. The specificity and selectivity of these biocatalysts provide the possibility of working under mild reaction conditions without generating unwanted side products and obtaining the product with a high degree of purity. Initially, polyester synthesis was catalyzed by enzymes in various organic solvents using a lower temperature range than used in organic chemistry in order to achieve high conversions and polyesters with high molecular weight [15–31].

Various studies on polycondensation have been performed using several lipases over the years [16,20–27]. Polymerization of 1,4-butanediol with adipic or sebacic acid has effectively been carried out in organic solvents using a free and immobilized lipase from *Mucor miehei* [28,29]. Enzymatic polymerization of dicarboxylic acids and glycols in a solvent-free system using *Candida antarctica* lipase B (CALB) immobilized as *Novozym 435*® could better tune the polymer structure and characteristics than the corresponding conventional chemical catalytic process [24,30,31].

Duwensee (2010) [16] studied the polycondensation of aliphatic diacids and diols catalyzed with CALB in free and immobilized preparations in a biphasic system (buffer/an organic solvent) using an integrated product removal–organic extraction phase, obtaining the polyester with 98% of conversion. *Pseudomonas cepacia* lipases have also been revealed to be useful for the polycondensation of sebacic acid with 1,8-octanediol in an aqueous medium [32].

The most frequent enzymatic polyester synthesis strategies in non-aqueous media are ring-opening polymerization (ROP) [33], solvent-free polymerization (bulk polymerization) [30], and direct polycondensation [24,25,31].

By direct polycondensation of dicarboxylic acids or their derivatives (hydroxyacids, esters) or diols (mainly dialcohols), more complex and structurally diverse monomers can be employed in comparison to enzymatic ROP [33]. However, those enzymatic polyester syntheses continue to operate under unfavorable polymerization conditions, such as the ones resulting from using organic solvents and high temperatures, or even when a low vacuum is used to remove the generated by-products, which may denature or inhibit the enzyme [32].

The nature of the polymerization medium and reaction conditions significantly influence the polyester characteristics and biocatalyst stability [15,34–36]. Compared to organic solvent media, the use of water as a solvent in polymerization has economic and environmental benefits. Kobayashi's research group [32] tested for the first time pure water for the polycondensation of several dicarboxylic acids and dialcohols at low concentration (0.1 M). These authors proved that polymerization in water using free lipases is possible. However, low solubility of polymer and monomer in water, enzyme and polymer separation challenge, and the use of organic solvents for polymer recovery from aqueous polymerization media were disadvantages related to those processes.

Aqueous miniemulsions (oil-in-water (*o/w*)) are systems characterized by oil droplets in the range between 200 and 300 nm, dispersed in a continuous water phase, which contributes to about 80% (*w/w*) [37]. Several reactions have been carried out in miniemulsion systems using enzymes by our research group, such as high-added-value ester synthesis using various lipases and cutinases [38–40], resolution of enantiomers with *Candida antarctica* lipase B (CALB) [41], triglyceride hydrolysis, and glycerolysis with *Pseudomonas cepacia* [42]. More recently, aldol reactions were performed using *Rhizopus arrhizus* lipase [43]. Additionally, aqueous miniemulsions have been reported as environmentally benign enzymatic reaction media for polymerization [37,44]. In these polymerization media, there is no need to dry reagents before the polymerization, the water produced in dehydrative polyesterification is expelled to the continuous (aqueous) phase, and the biocatalyst stability is improved in comparison to an organic solvent. Moreover, there is more efficient product separation [44]. The acid-catalyzed polyesterification in the miniemulsion process allows the synthesis of hydrophobic polyesters under very favorable conditions such as low temperature [11,12].

Enzyme stability and polymer characteristics also depend on the type of reactor and process conditions such as type of biocatalyst, temperature, pH, and type of stirring [37,44]. The polymerization processes can be carried out in batch and continuous reactors. Continuous processes are used for large-scale production, while batch and fed-batch operation modes are used typically for low- and high-value products [45]. Batch processes are more suitable for enzymatic polymerizations, where they may have superior selectivity [46,47].

With this work, we particularly want to expand the understanding of the lipase-catalyzed polycondensation of diacids with diols in o/w miniemulsion systems and compare them with the organic solvent (cyclohexane:tetrahydrofuran (THF) 5:1 *v/v*) and a pure aqueous medium. The present work also investigated the formation of poly(octamethylene suberate) (POS) by using miniemulsion systems. The influence of several parameters such as type of lipase, free and immobilized lipase specificity and selectivity, pH, and temperature on the polymer molecular weight was studied.

Regarding the importance of viscosity for polycondensation, special attention was focused on the evaluation of stirring conditions (type, rate) in correlation with other process parameters such as temperature and pH. As a high concentration of substrates used in the polymerization medium can influence the polymer molecular weight, substrate conversion, enzyme inhibition, and stability, the design of a fed-batch substrate supply was tested. Finally, scale-up in a small reactor with an impeller (RI) and immobilized enzyme reutilization in different polymerization media were compared. Furthermore, the final quality of the synthesized poly(octamethylene suberate) (POS) was characterized.

Lipase-catalyzed polycondensation in a renewed aqueous polycondensation medium, as part of the polymer-5B technology, was applied in this work to synthesize biodegradable and biocompatible polyesters with desirable polymer characteristics and good monomer conversion [48].

## 2. Materials and Methods

### 2.1. Enzymes and Chemicals

The free and immobilized enzyme preparations tested in this work were *Candida rugosa* lipase (CRL) and CRL immobilized on a synthetic support Immobead 150 (CRLI) (Sigma Aldrich, Darmstadt, Germany), *Burkholderia cepacian* (BCL) (also named Amano Lipase PS) and BCL adsorbed in diatomite (BCLI) (Sigma Aldrich, Darmstadt, Germany), and *Pseudozyma antarctica* lipase B (PBL), previously known as *Candida antarctica* lipase B (CALB), and PBL adsorbed on an acrylic support (PBLI) (Chiral Vision, Den Hoorn, The Netherlands). Dicarboxylic acids such as adipic or hexanedioic acid (99%, Acros Organics, Geel, Belgium) and suberic or octanedioic acid (99%, Acros Organics, Geel, Belgium) and dialcohols such as 1,6-hexanediol (97%, Acros Organics, Geel, Belgium), 1,8-octanediol (98%, Acros Organics, Geel, Belgium), and 1,10-decanediol (98%, Sigma Aldrich, Darmstadt, Germany) were tested in polyester synthesis. Triton X-100 (Merck KGaA,

Darmstadt, Germany) was used to obtain stable *o/w* miniemulsion systems. Cyclohexane (99.5%, Merck Darmstadt, Germany) and tetrahydrofuran (THF) (99%, with 250 ppm BHT as an inhibitor; Honeywell, Charlotte, NC, USA) were mixed and constituted the organic solvent tested for comparison in polyester synthesis.

All reagents were used without any additional modification, except 1,8-octanediol, which was reduced to a more fine-size powder in a mortar to obtain better dispersion in the aqueous polycondensation media. The beads of the different immobilized enzyme preparations were previously washed with Milli-Q water (Merck Millipore, Darmstadt, Germany), as indicated by the suppliers, and dried at room temperature before use.

*2.2. Methods*

2.2.1. Activity of the Free and Immobilized Enzyme Preparations

An activity assay was performed using a titration method of the hydrolysis of tributyrin (TBU, expressed as μmol of tributyrin hydrolyzed per minute) to confirm the activity and function of free and immobilized enzyme preparations. This method used an *o/w* emulsion system containing 30 mM of tributyrin (98%, Sigma Aldrich, Darmstadt, Germany), 100 mM of NaCl (99.5%, Applichem Panreac, Barcelona, Spain), 3.5% (*v/v*) of Triton X-100 (Merck KGaA, Darmstadt, Germany), and sodium phosphate buffer (25 mM, pH 8.0) at 30 °C. The release of butyric acid was automatically titrated with the alkaline reagent (NaOH solution) added by the Methrom Titrino 702 SM syringe. The pH was kept constant at 8.0 with standard NaOH solution.

The activity (Act) of enzyme preparations was expressed in μmol of butyric acid released per gram of biocatalyst and per minute and determined by the following Equation (1):

$$Act = \frac{\frac{\Delta v}{\Delta t} \cdot M}{m} \tag{1}$$

where $\Delta V_{NaOH}/\Delta t$ (mL NaOH min$^{-1}$) is the slope between the volume of the standard NaOH solution added and consumed to keep the pH constant and equal to 8 as a function of hydrolysis time, M is the NaOH molarity of the titration solution (mM), and m is the mass of the biocatalyst (g) of the powder formulation of the free enzyme or gram of support with the immobilized enzyme, both on a dry basis. The results are presented in Table 1.

**Table 1.** The activity of free and immobilized enzyme preparations assayed according to the titration method of the hydrolysis of tributyrin.

| Biocatalysts | Activity |
|---|---|
| | μmol Butyric Acid (g min)$^{-1}$ |
| Free *Candida rugosa* lipase (CRL) | 1368 |
| Immobilized *Candida rugosa* lipase (CRLI) | 84 |
| Free Lipase *Burkholderia cepacia* lipase (BCL) | 10,140 |
| Immobilized *Burkholderia cepacia* lipase (BCLI) | 14,100 |
| Free *Pseudozyma antarctica* lipase B (PBL) | 5478 |
| Immobilized *Pseudozyma antarctica* lipase B (PBLI) | 1224 |

Biocatalyst activity was also assayed during polycondensation in miniemulsion using Equation (1) but substituting *m* by the volume of the sample withdrawn in the beginning ($A_o$) and at the end of the polymerization (*A*). The activity was expressed in μmol of butyric acid released per mL and per minute.

2.2.2. Miniemulsion Preparation

Relative amounts of equimolar substrate concentrations (16.8%) (dicarboxylic acids and dialcohols), surfactant (1.6%) (Triton X-100), and water (81.6%) that made up the miniemulsion used as the polycondensation medium were thoroughly mixed and homogenized by magnetic stirring for 1 h (500 rpm, 25 °C). The two-phase system was

ultrasonicated for 120 s in pulses of 5 s and pauses of 10 s at 50% amplitude (SONOPLUS, Bandelin, tip MS72), obtaining a milky emulsion [46].

### 2.2.3. Polyester Synthesis in Miniemulsion

The enzymatic polyester synthesis was performed in miniemulsion using one diacid (hexanedioic or octanedioic acid) and one diol (1,6-hexanediol, 1,8-octanediol, or 1,10-decanediol) at an equimolar concentration. If not otherwise stated, the equimolar monomer concentration was 0.5 M, with an initial pH of 3.3 in the miniemulsion. For some experiments, pH was corrected to 4.0, 5.0, or 6.0 using 0.5 M NaOH solution. The biocatalysts tested for the polycondensation were free (CRL, BCL, PBL) and immobilized (CRLI, BCLI, and PBLI) lipase preparations (Table 1).

The polyester synthesis was carried out in 20 mL capped flasks used as a reactor at 25, 35, 45, or 65 °C under direct magnetic stirring (250 rpm) (reactor with magnetic stirring (RMS)) for 48 h, and 10 mL of the miniemulsion was added to the reaction vessel containing the appropriate amount of free enzymes or immobilized enzymes (5 or 8 mg mL$^{-1}$, respectively).

Samples were withdrawn directly from the polycondensation medium under high homogenization to determine substrate conversion (%) (Section 2.2.6) and the polymer molecular weight (Section 2.2.7.1).

### 2.2.4. Poly(octamethylene suberate) (POS) Synthesis with the PBLI Biocatalyst
### 2.2.4.1. In Batch Operation Mode

Poly(octamethylene suberate) (POS) synthesis was performed between octanedioic acid (suberic acid) and 1,8-octanediol at an equimolar concentration (0.5 M) in three different polycondensation media: miniemulsion, water, and an organic solvent system comprising a mixture of cyclohexane and THF (5:1 $v/v$). The initial pH of the miniemulsion and water was 3.3 and in other experiments was corrected to 5.0 using 0.5 M NaOH solution.

The POS synthesis was carried out in 20 mL capped flasks used as the reactor (RMS) at 45 °C under direct magnetic stirring (250 or 500 rpm) for 48 h unless otherwise stated in the text. The polycondensation in batch operation mode started with the addition of the PBLI biocatalyst (8 mg mL$^{-1}$). Samples were withdrawn to determine substrate conversion (%) (Section 2.2.6) and the polymer molecular weight (Section 2.2.7.1).

### 2.2.4.2. In Fed-Batch Operation Mode

According to fed-batch operation mode, the POS synthesis was performed equally for the three polycondensation media (miniemulsion, water, and the organic solvent). The POS synthesis was carried out in a 20 mL glass vial with an initial volume of 10 mL, under direct magnetic stirring (RMS) at 250 rpm and a temperature of 45 °C. The pH correction to 5.0 was done only in the miniemulsion and water using a 0.5 M NaOH solution. The polymerization started with the addition of PBLI (8 mg mL$^{-1}$).

The fed-batch protocol is presented in Figure 1. The reaction started with using a 0.1 M equimolar concentration of monomers (1,8-octanediol and octanedioic acid). Then, consecutive additions equivalent to 0.1 M of monomers every hour were made, up to a total of both monomers' concentration of 0.5 M.

After 1 h, the reaction at the first reactor (R1) was ended, and the conversion (%) and polymer molecular weight (g mol$^{-1}$) were assayed. Simultaneously (after 1 h), 1,8-octanediol and octanedioic acid were added into reactors R2, R3, R4, and R5 to enhance the concentration of both monomers equivalent to 0.1 M in each polycondensation medium. This protocol was repeated for reactors R3, R4, and R5, each time increasing the monomer concentration (0.1 M) according to Figure 1. The polycondensation was carried out for 1 h, 2 h, 3 h, and 4 h for R1, R2, R3, and R4, respectively. For reactor R5, after achieving a substrate concentration of 0.5 M, a sample assay was performed at 5 h and 48 h.

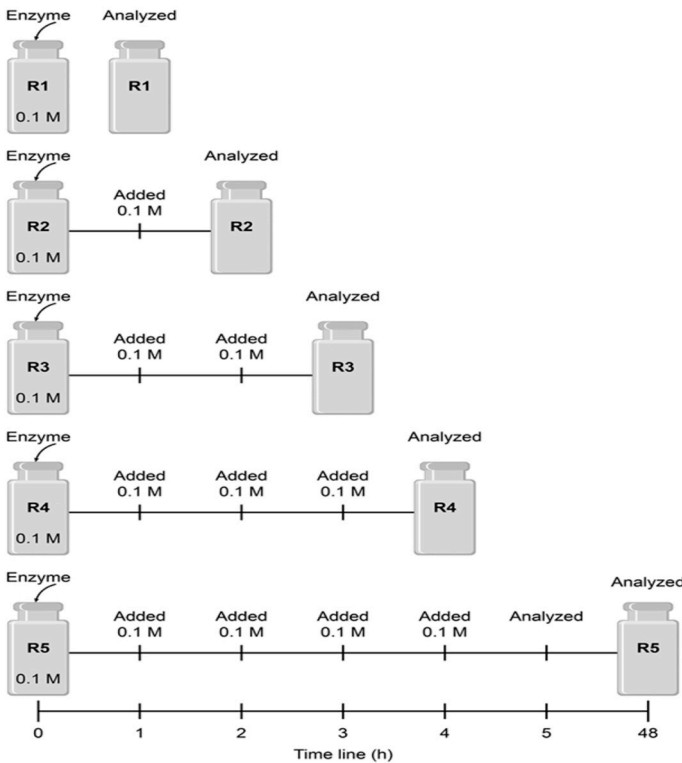

**Figure 1.** Schematic presentation of the fed-batch operation mode used on the poly(octamethylene suberate) (POS) synthesis for the three polycondensation media (miniemulsion, water, and the organic solvent) at 45 °C, pH 5.0, and 250 rpm.

### 2.2.4.3. Effect of the Stirring Type

The POS synthesis was performed in batch operation mode using plastic vessel in Teflon (polytetrafluoretileno—PTFE) (miniemulsion and water) or metallic (organic solvent) reactors of 70 mL with an impeller (RI) constituted by a magnetic stirrer rod (with a two-blade propeller). The impeller was immersed in 50 mL of polycondensation media with an equimolar substrate concentration (0.5 M). The small reactor (10 mL) with direct magnetic stirring (RMS) was used for comparison. The initial pH correction to 5.0 was carried out in the miniemulsion and water using 0.5 M NaOH. The reactors were kept inside a thermal incubator (AGITORB 160 E, Aralab, Lisbon, Portugal) at 45 °C. The polycondensation was catalyzed by PBLI (8 mg mL$^{-1}$) at 250 rpm for 8 h. The second group of experiments in the RI was tested at 500 rpm and a temperature of 60 °C. The conversion (%) and polymer molecular weight (g mol$^{-1}$) of the polycondensation were assayed.

### 2.2.5. Biocatalyst Reutilization (PBLI) in a Reactor with an Impeller (RI)

The PBLI reutilization (8 mg mL$^{-1}$) was tested for the POS synthesis in the three polymerization media (miniemulsion, water, and the organic solvent) using the RI in batch operation mode. The reaction conditions have been explained previously (Section 2.2.4.3), but in this experiment, a filter constituted by a stainless steel network (70 mesh) was fixed at the bottom of the reactor (RI). Each polycondensation cycle was carried out for 2 h at 60 °C and 500 rpm. After 2 h of polymerization (1st cycle), each polycondensation medium (50 mL) was filtrated into a new container with the aid of a peristaltic pump (Easy-load MasterFlex, Model 7518-00, Vernon, IL, USA), while the stainless steel network retained the PBLI biocatalyst. Then, 50 mL of water (for the miniemulsion and water polymerization media) and organic solvent, previously heated at 60 °C, were added into the respective reactor (RI) to wash the biocatalyst. The washed biocatalyst suspension was filtered quickly again before starting a new cycle. Then, 50 mL of each polycondensation medium (previously heated at 60 °C) was added to the reactor containing the washed

biocatalyst and then the 2nd polycondensation cycle was started. The procedure was repeated seven times (i.e., a total of 7 cycles). For the 6th cycle, the polycondensation was evaluated after 24 h and for the 7th cycle after 48 h of polycondensation.

The conversion (%) by titration of free acids and the polymer molecular weight (g mol$^{-1}$) of POS in the corresponding filtrate of each polycondensation medium were assayed.

### 2.2.6. Determination of Acidity and Conversion

The conversion (%) was calculated using the initial and final analysis of the acidity expressed in mg KOH g$^{-1}$ in each polycondensation medium, according to the AOCS Official Method Te1a-64. The remaining acid titration with standard KOH solution (0.5 M) was performed in triplicate using the Methrom Titrino 702 SM.

The samples withdrawn directly from the polycondensation media or filtrates were weighed and diluted in 5 mL of ethanol, and the acidity and conversion of dicarboxylic acid were calculated using the following equations:

$$Acidity \ (\mathrm{mg \ KOH/g}) = \frac{V \cdot M \cdot MW(KOH)}{m} \quad (2)$$

$$Conversion \ (\%) = \frac{Initial \ acidity - final \ acidity}{Initial \ acidity} \times 100 \quad (3)$$

where $V$ is the spending volume of KOH solution in mL, M is the molarity of KOH solution in $M$, $m$ is the weight of the sample (g), and MW (KOH) is the molecular weight of KOH (56.1 g mol$^{-1}$).

### 2.2.7. Polymer Characterization

The molecular weights of the samples withdrawn from polycondensation media or filtrates were analyzed by size-exclusion chromatography, nuclear magnetic resonance spectroscopy, thermogravimetric analysis, and differential scanning calorimetry analysis.

#### 2.2.7.1. Polyester Molecular Weight Assay by Size-Exclusion Chromatography (SEC)

The average molecular weight ($M_w$) of the polyesters was determined by size-exclusion chromatography (SEC) using a high-performance liquid chromatography (LaChrom HPLC) apparatus equipped with a refractive index detector (Merck LaChrom RI Detector L-7490) and a polystyrene/polydivinylbenzene column (ResiPore Agilent). The elution solvent was THF at a flow rate of 0.5 mL min$^{-1}$ at 40 °C. The calibration curve with polystyrene standards of molecular weight between 660 and 482,400 g mol$^{-1}$ was determined. The miniemulsion and water samples were centrifugated for 10 min at room temperature (25 °C) and 10,000× $g$ (Eppendorf Centrifuge 5415 D), removing the water supernatant. The polymer and monomers precipitated from the miniemulsion and water were washed and dried in an oven (Memmert). The organic solvent samples were evaporated with precipitation of the polymer and monomers and dried in an oven (Memmert, GmbH, Schwabach, Germany).

The standards and dried samples were solubilized in THF, submitted to a thermal shock at 40 °C for 5 min, and then centrifuged at room temperature before SEC analysis. The experimental error associated with repeated injection of the same polymer sample was inferior to 3%.

#### 2.2.7.2. Nuclear Magnetic Resonance ($^1$H NMR) Spectroscopy

The molecular structure of the synthesized polyester, namely POS, was confirmed by $^1$H NMR. $^1$H NMR spectroscopy analysis was performed on a Bruker NMR nuclear magnetic resonance spectrometer, operating at 300 MHz using 5-mm-internal-diameter tubes. The solvent using was deuterated chloroform (CDCl$_3$-d) (99.8%, Cambridge Isotope Laboratories) at a concentration of 6 mg mL$^{-1}$. A relaxation delay of 2 s was used with a total of 64 scans. The spectra were compared regarding the residual CDCl$_3$-d peak

(at 7.3 ppm), and the chemical shifts ($\delta$) were related in parts per million (ppm) to the chloroform solvent and integrated using Bruker software.

#### 2.2.7.3. Thermogravimetric Analysis (TGA)

Through thermogravimetric analysis (TGA 92-16.18 Setaram), the thermal stability, melting temperature ($T_m$), and maximum weight loss were assayed. The heating ramp was 10 °C min$^{-1}$, with the temperature varying between 20 and 600 °C. The analysis was performed under a nitrogen atmosphere (60 mL min$^{-1}$).

#### 2.2.7.4. Differential Scanning Calorimetry (DSC) Analysis

Differential scanning calorimetry (DSC) analysis was used to confirm the polymer melting temperature that typically correlates directly with the molecular weight, the repeating-unit carbon chain length of the polymer, and some impurity.

DSC analysis was performed in a 2920 MDSC System (TA Instruments Inc., New Castle, DE, USA). For this analysis, 5 to 6 mg of dry polyester, particularly poly(octylmethylene suberate), was weighed in a standard aluminum cuvette, while another empty cuvette was used as a reference, and both were thermally sealed. Each sample was subjected to a heating cycle of −60 °C to 110 °C at a rate of 5 °C min$^{-1}$. Nitrogen at a flow rate of 60 mL min$^{-1}$ purged and inert the DSC cell atmosphere. The fusion points ($T_f$) and enthalpies ($\Delta$) using appropriate equipment software were evaluated.

### 3. Results and Discussion

#### 3.1. Polyester Synthesis in the Miniemulsion

The influence of three different lipases, in free (CRL, BCL, PBL) and immobilized (CRLI, BCLI, and PBLI) forms, on the polyester synthesis from different diacids and diols in a miniemulsion was evaluated.

Initially, an equimolar monomer concentration of 0.5 M of octanedioic acid and 1,8-octanediol was tested for the POS synthesis with free lipases (CRL, BCL, PBL) (Figure 2, Table 2). The free lipases present different catalytic efficiencies, mainly dependent on the initial pH and temperature.

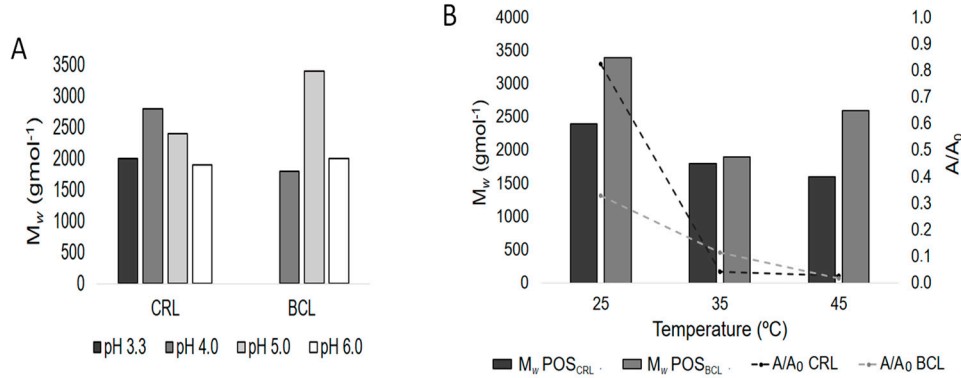

**Figure 2.** POS molecular weight ($M_w$) and relative activity ($A/A_0$) in a miniemulsion of CRL and BCL at different initial pH values (3.3, 4.0, 5.0, 6.0) at 25 °C (**A**) and different temperatures (25 °C, 35 °C, 45 °C) for CRL and BCL at pH 4.0 and 5.0, respectively (**B**). Reaction conditions: equimolar monomer (1,8-octanediol and octanedioic acid) concentration (0.5 M); 250 rpm; enzyme: 5 mg mL$^{-1}$ of free enzyme preparations; reaction time 48 h.

Enzyme activity depends on the pH of the reaction media, which in the case of a miniemulsion system is determined by the continuous (water) phase [40]. A comparison of the polymer molecular weight at pH 3.3 (pH of the original miniemulsion) and miniemulsions with the corrected pH value was made (Figure 2A and Table 2). A higher POS molecular weight and activity were obtained for CRL at pH 4 (2750 g mol$^{-1}$) and for BCL (3250 g mol$^{-1}$) and PBL (6950 g mol$^{-1}$) at pH 5 at 25 °C (Figure 2B and Table 2). POS formation catalyzed with BCL at pH 3.3 was not observed (Figure 2A).

**Table 2.** POS molecular weight ($M_w$) obtained by the polycondensation in the miniemulsion catalyzed by PBL and PBLI at different pH and temperature.

| pH | Temperature (°C) | $M_w$ (g mol$^{-1}$) | |
| | | PBL | PBLI |
|---|---|---|---|
| 3.3 * | 25 | 2600 | 2700 |
| 5 | 25 | 6950 | 7300 |
| 3.3 * | 45 | 2600 | 3450 |
| 5 | 45 | 3550 | 7600 |

* Initial pH of the miniemulsion without correction. Polyester synthesis conditions: 1,8-octanediol and octanedioic acid as monomers at an equimolar concentration (0.5 M), 250 rpm; 5 mg mL$^{-1}$ of PBL and 8 mg mL$^{-1}$ of PBLI; and 48 h of polycondensation.

The effect of temperature on lipase activity was evaluated in the range of 25–45 °C at optimum pH previously observed for each enzyme (CRL at pH 4; BCL and PBL at pH 5). The CRL and BCL activities decreased significantly at temperatures higher than 30 °C, obtaining the highest molecular weight of 2300 and 3400 g mol$^{-1}$, respectively, at 25 °C (Figure 2B). These results were due to the loss of the enzyme secondary structure with a temperature increase and a parallel decrease in activity in the polycondensation media [49]. However, no variation in the molecular weight (2600 g mol$^{-1}$) was observed by a temperature change at pH 3.3 for PBL (Table 2). An inhibition effect of temperature on PBL at 45 °C ($M_w$ of 3550 g mol$^{-1}$) in comparison to 25 °C ($M_w$ of 6950 g mol$^{-1}$) was observed for a pH value of 5.0 (Table 2).

Due to the lower catalytic efficiency of CRL and BLC than PBL in the POS synthesis at 0.5 M monomer concentration, the free enzymes (CRL, BCL, and PBL) were tested for different equimolar monomer (1,8-octanediol and octanedioic acid) concentrations in the range from 0.05 to 0.6 M at 25 °C and 48 h of polycondensation. The maximum POS molecular weight of 3550 g mol$^{-1}$ for CRL was observed at a concentration of 0.1 M. In the case of BCL and PBL, the maximum POS molecular weight of 3850 and 6950 g mol$^{-1}$, respectively, was detected at a concentration of 0.5 M. Based on these data, an increase in the monomer concentration above 0.1 M showed a high inhibition effect on CRL. Besides the maximum polymer weight observed for the same monomer concentration (0.5 M) for BCL and PBL, PBL presented higher monomer selectivity with regard to the higher molecular weight obtained.

The respective immobilized enzyme preparations (CRLI, BCLI, and PBLI) were also tested for the same initial polycondensation conditions in the miniemulsion. The three immobilized lipase preparations showed different behaviors in comparison to the free enzymes. The POS synthesis using CLRI and BCLI did not occur, while PBLI catalytic activity in POS synthesis was more efficient than the free enzyme (PBL) (Table 2). The POS molecular weight obtained by PBLI at pH 3.3 (2700 g mol$^{-1}$ for 25 °C and 3450 g mol$^{-1}$ for 45 °C) was slightly higher than the POS molecular weight obtained by PBL at the same pH (3.3) (2600 g mol$^{-1}$ for both temperatures). POS synthesis at pH 5 led to significant molecular weight differences, especially at 45 °C (i.e., 3550 and 7600 g mol$^{-1}$ for PBL and PBLI, respectively) (Table 2).

The lack of polycondensation with immobilized enzymes, CRLI and BCLI, can be due to the size of the miniemulsion droplets that are unable to penetrate inside the support pores where the immobilized lipases (CRLI and BCLI) are localized. The other possibility is that the lipases immobilized in CRLI and BCLI biocatalysts could not successfully perform the interface activation mechanism characteristic of these specific lipases. This hypothesis could be the more plausive explanation as PBLI presented excellent results (7600 g mol$^{-1}$, 45 °C), and PBL is an atypical lipase not characterized by the interface activation mechanism, according to several authors [50]. Additionally, the high catalytic efficiency of PBLI to catalyze polyester synthesis in a miniemulsion may also be related to the specific support–enzyme interactions already observed for a similar biocatalyst named Novozym 435 and the intrinsic stability characteristics of PBL [51].

The initial pH correction from pH 3.3 to pH 5.0 in the miniemulsion for free and immobilized enzyme preparations (PBL and PBLI) at 25 °C increased the polymer molecular weight 2.7-fold. However, with the same pH correction at 45 °C, the polymer molecular weight increased only 1.4- and 2.2-fold for free (PBL) and immobilized (PBLI) enzyme preparations, respectively (Table 2). These results confirm higher activity and probably stability of the immobilized enzyme preparation (PBLI) for polycondensation at 45 °C in miniemulsion systems compared to the free enzyme (PBL).

Barrère et al. (2003) [11], by direct dehydration polycondensation of dodecanediol with dodecanedioic acid or *tert*-butyl-3-adipic acid at 70 °C for 100 h in an emulsion system containing 16 wt % of *p*-dodecylbenzene sulphonic acid (DBSA), obtained a conversion yield of 85% and a molecular weight of 3480 or 1100 g mol$^{-1}$, respectively. Takasu (2006) [12], using a similar emulsion system with the DBSA catalyst, obtained 10100 g mol$^{-1}$ and excellent yield (99%) at higher polymerization temperature (85 °C) for 48 h. However, in this last work, the direct dehydration polycondensation occurred concomitantly with chain extension using hexamethylene diisocyanate as the chain extender. In our work, it was possible to run the polycondensation at a lower temperature (25 or 45 °C) and also easily separate the biocatalyst (PBLI) in comparison to a soluble catalyst, like DBSA [11,12].

The effect of the acid and alcohol carbon chain length on polyester synthesis by PBLI in a miniemulsion was assessed for hexanedionic (C$_6$) and octanedioic (C$_8$) diacids, while the diol carbon chain length increased from C$_6$ to C$_{10}$ atoms for both diacids.

The polycondensation was carried out at equimolar monomer concentrations (0.5 M) in the miniemulsion, at initial pH 5.0, 250 rpm, and 45 °C, for 48 h using 8 mg mL$^{-1}$ of PBLI (Figure 3).

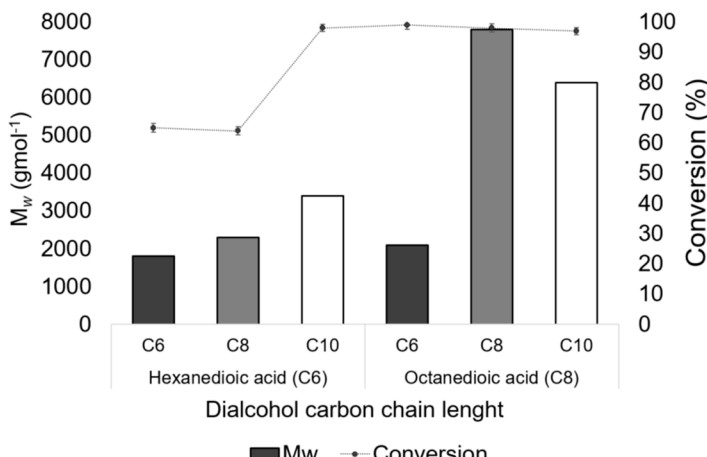

**Figure 3.** Polyester molecular weight ($M_w$) and conversion of polycondensation in a miniemulsion with PBLI (8 mg mL$^{-1}$) using equimolar concentrations (0.5 M) of monomers with different carbon chain lengths at pH 5.0 and 250 rpm.

The diacid carbon chain length showed an evident influence on the molecular weight of the polymer, except for the C$_6$ dialcohol. By increasing the carbon chain length of dialcohol (C$_6$ to C$_{10}$) in polycondensation with hexanedioic acid (C$_6$), the polymer molecular weight gradually increased from 1900 g mol$^{-1}$ (C$_6$) to 3300 g mol$^{-1}$ (C$_{10}$). Furthermore, high conversion (>98%) for monomers with the longer carbon chain length of dialcohol (C$_{10}$) with hexanedioic acid (C$_6$) was also observed (Figure 3). Enzyme stabilization with the dicarboxylic acids in opposition to the destabilization induced by the dialcohols in miniemulsion systems was previously observed [49].

For octanedioic acid, with an increasing carbon chain length of dialcohol from C$_6$ to C$_8$, a significant increase in polymer molecular weight from 2100 to 7800 g mol$^{-1}$ was observed. However, using the dialcohol C$_{10}$, the polymer molecular weight decreased by 45% (from 7800 to 4300 g mol$^{-1}$ for C$_8$ and C$_{10}$, respectively) (Figure 3). The results demonstrated that PBLI mostly shows higher specificity for longer carbon chains of dicarboxylic acids

and dialcohols under the experimental conditions tested (Figure 3). PBLI showed higher substrate specificity for the $C_8$ carbon chain length of the dicarboxylic acid and dialcohol in polyester synthesis in the miniemulsion. *Candida antarctica* lipase B (CALB) has been reported to present substrate specificity [50,51] and high selectivity for substrates with a longer alcohol carbon chain [52].

Duwensee et al. (2010) [16] performed polycondensation of sebacic acid and 1,4-butanediol catalyzed in a biphasic system comprising citric acid/phosphate buffer and an organic solvent diisopropyl ether (DIPE) or *tert*-butyl methyl ether (MTBE) using CALB in free and immobilized forms. At the end of the polycondensation at 50 °C, for substrate concentrations of 0.2 M, at pH 6, and after 48 h of polycondensation, molecular weights of 1890 or 2520 g mol$^{-1}$, respectively, were detected.

Due to the high specificity shown by PBLI for dicarboxylic acids and dialcohols with a carbon chain length of $C_8$, these monomers were chosen in this work to analyze the effect of reactor conditions on polyester synthesis.

High conversion values (>98%) were achieved for the dicarboxylic acid $C_8$ (octanedioic acid) with an increasing chain length of dialcohols ($C_6$, $C_8$, and $C_{10}$) (Figure 3). This high conversion was confirmed by the only traces of monomers detected by SEC analysis illustrated in Figure 4. The monomers' (octanedioic acid and 1,8-octanediol) peak presents a residence time of 22.6 min corresponding to an area of 3421201 at time zero (Figure 4A), and no significant peak was detected by SEC after 48 h of polycondensation (Figure 4B), confirming that conversion of about 100% occurred.

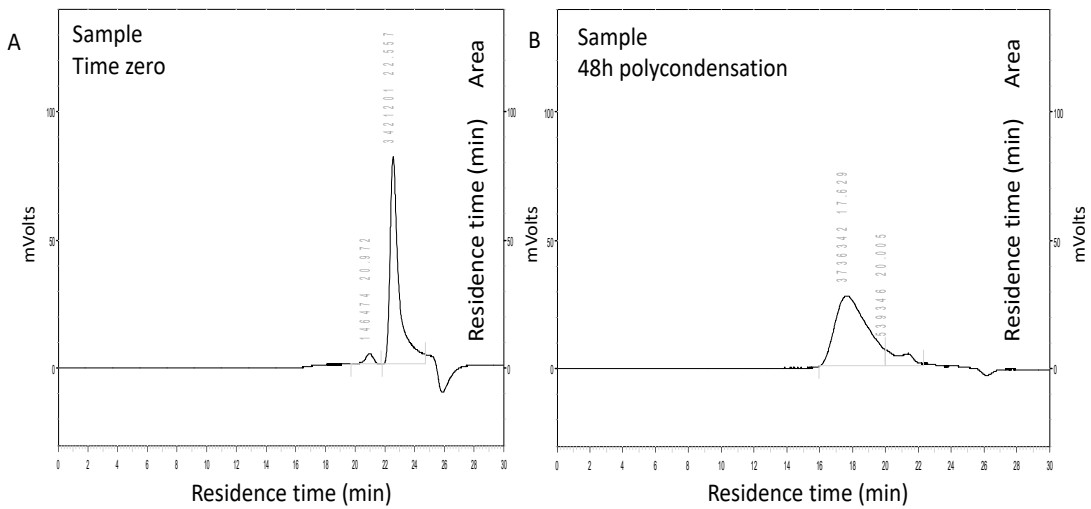

**Figure 4.** Size-exclusion chromatography (SEC) spectra obtained from the synthesis of poly(octylmethylene suberate). (**A**) Sample at time zero with an average $M_w$ = 180 g mol$^{-1}$ for residence time = 22.6 min with an area of 3421201, corresponding to a mixture of octanedioic acid and 1,8-octanediol for equimolar monomer concentrations (0.5 M). (**B**) Sample after 48 h of polycondensation at 45 °C with synthesized polyesters corresponding to residence time = 17.6 min with average $M_w$ = 7800 g mol$^{-1}$, and SEC detected no significant peak for 22.6 min.

### 3.2. Effect of Reactor Conditions on POS Synthesis in the Miniemulsion

When considering the properties of polymers prepared by emulsion polymerization, the reactor conditions have a critical impact. The viscosity of the monomer/polymer solution increases fast, with monomer conversion influencing colloidal stability and monomer–monomer and enzyme–monomer interactions, so efficient mixing is essential to obtaining good polymer properties. The polymerization media must be stirred continuously during the polycondensation using a suitable agitator design and speed [37,45].

The effects of the miniemulsion preparation and composition and stirring conditions were evaluated on the POS synthesis at equimolar concentrations (0.5 M) of 1,8-octanediol and octanedioic acid for 48 h, catalyzed by the PBLI biocatalyst (8 mg mL$^{-1}$) (Table 3).

The polymer molecular weight and conversion (%) obtained under these conditions are illustrated in Table 3 and Figure 5.

**Table 3.** Specifications of temperature, pH, and stirring in POS synthesis in the miniemulsion using PBLI (8 mg mL$^{-1}$) and respective conversion achieved under each experimental polycondensation condition after 48 h.

| Experiment Condition | Temperature (°C) | pH | Stirring (rpm) | Polycondensation System | Conversion (%) |
|---|---|---|---|---|---|
| A | 45 | 3.3 * | Magnetic, 500 | Miniemulsion | 92 ± 0.2 |
| B | 45 | 5 | Magnetic, 500 | Miniemulsion | 99 ± 0.4 |
| C | 45 | 5 | Magnetic, 250 | Miniemulsion | 98 ± 0.2 |
| D | 45 | 5 | Orbital, 250 | Miniemulsion | 99 ± 0.6 |
| E | 25 | 5 | Magnetic, 250 | Miniemulsion | 99 ± 0.2 |
| F | 65 | 5 | Magnetic, 250 | Miniemulsion | 91 ± 0.4 |
| G | 45 | 5 | Magnetic, 250 | Emsulsion without sonication | 97 ± 1.2 |
| H | 45 | 3.3 * | Magnetic, 250 | Water | 94 ± 0.3 |

* pH of the polycondensation reaction media, without correction with NaOH solution.

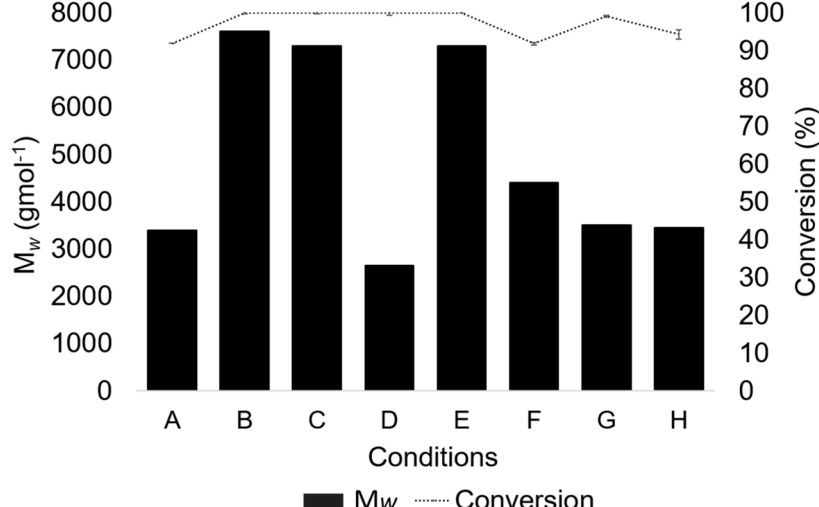

**Figure 5.** POS molecular weight ($M_w$) and conversion after 48 h of polycondensation in the miniemulsion under different parameter conditions (Table 3) using PBLI (8 mg mL$^{-1}$) as the biocatalyst.

An increase in the direct magnetic stirring to 500 rpm for initial pH 3.3 and pH 5.0 (conditions A and B, Table 3 and Figure 5) at 45 °C led to a polymer molecular weight of 3500 and 7800 g mol$^{-1}$, respectively, and under both pH conditions, with conversion higher than 90% (Table 3). The pH influence on the polymer molecular weight was already observed at 250 rpm (Table 2). The increase in direct magnetic stirring from 250 rpm to 500 rpm in the miniemulsion showed a relatively low polymer molecular weight increase at 45 °C (7.7%, conditions B and C, Table 3 and Figure 5). Additionally, at 500 rpm, formation of agglomerates occurred, containing probably monomers, the polymer, and biocatalyst, glued to the magnetic stirrer and the reactor wall at the end of the polymerization. This singularity was not observed in the experiment carried out at direct magnetic stirring at 250 rpm.

For this reason, the polycondensation was also tested in an orbital shaker at 250 rpm (condition D, Table 3 and Figure 5). In the orbital shaker, the PBLI beads were almost retained in the bottom of the reactor with low dispersion and homogenization, and additionally, there was no agglomerate formation. The use of orbital stirring led to a 60% polymer molecular weight decrease compared to direct magnetic stirring at 250 rpm. Due to high external mass transfer resistance, less contact occurred between the biocatalyst and the polymer already formed, affecting the polymer growth (conditions C and D, Table 3

and Figure 5). Despite this, there was contact between the biocatalyst and the monomers as the conversion was 99%.

As an increase in magnetic stirring to 500 rpm showed a low influence on polymer molecular weight and conversion (%), further experiments were carried out at 250 rpm.

The viscosity dependence of temperature is an important property of polymerization systems, so the temperature of 65 °C was tested to analyze its influence on the POS molecular weight (condition F, Table 3 and Figure 5) compared to 25 °C and 45 °C (conditions E, C, Table 3 and Figure 5). At 25 and 45 °C, the polymer molecular weight change was less than 0.5%, but a critical drop in the polymer molecular weight from 7200 to 4300 g mol$^{-1}$ was detected by increasing the temperature from 45 to 65 °C for 48 h of polycondensation. Simultaneously, the conversion dropped from $98 \pm 0.2\%$ to $91 \pm 0.4\%$ (conditions C and F, Table 3 and Figure 5). The lower POS molecular weight and conversion degree observed at 65 °C could be due to the thermal inhibition of the enzyme activity of the PBLI biocatalyst.

Energy consumption is an important parameter for industrial application, so the polycondensation was tested for the emulsion system without the sonication step and the use of pure water as a polymerization medium (conditions G and H, Table 3 and Figure 5). Since miniemulsion droplets are like nanoreactors, their size and stability can have a significant impact on the polycondensation performance in miniemulsions. A POS molecular weight of 7600 g mol$^{-1}$ was achieved in the miniemulsion, considerably higher than 3500 and 3650 g mol$^{-1}$ obtained for the emulsion without sonication and pure water, respectively, although in all the polycondensation media tested, the conversion was higher than 90%.

By applying ultrasonication on the emulsion formation, the droplet size decreased, increasing the droplet interfacial area available for enzyme action on monomers or the polymer already formed and accumulated inside miniemulsion droplets [11,36]. This phenomenon is confirmed by the lower molecular weights obtained in pure water and the emulsion without ultrasonication.

On the other hand, the POS synthesis in water also proves that PBLI can perform polycondensation without the emulsion or organic solvent indispensable for promoting the interface activation mechanism of a typical lipase [50]. This result was surprising, in particular for pure water, as there was a considerable amount of monomers that are insoluble in water and remain initially in solid form (precipitate). The same precipitation behavior in water is expected for POS synthesized due to its high hydrophobicity and low water solubility. The polycondensation results obtained in pure water suggest that there is a small fraction of monomers and polyester soluble and in equilibrium with respective precipitates in pure water. The monomers and polyester gradually dissolved and diffused into the active site of the immobilized enzyme (PBLI), where they were converted initially to small-chain-length polyesters but increasing with polymerization time. Thus, it is possible to classify this as polycondensation from solid to solid via biocatalysis in water [53].

The best performance regarding the POS molecular weight and substrate conversion was accomplished at 45 °C with magnetic stirring at 250 rpm in the miniemulsion.

*3.3. Effect of the Reaction Operation Mode: Batch vs. Fed-Batch for Different Polymerization Media*

The enzymatic POS synthesis was carried out in the previous subsection in batch operation mode with the initial concentration of 0.5 M of each monomer (1,8-octanediol and octanedioic acid). Polycondensation in batch operation mode allows flexibility and adaptability to obtain different polyesters [54]. However, the inhibition effect of the high substrate concentrations on lipase activity in the miniemulsion, as well as in an organic solvent, was previously observed for the esterification [38–40,55]. To overcome the inhibition effect of high substrate concentration on enzyme activity, the reactions can be carried out in fed-batch operation mode, which means the stepwise addition of substrates [55].

The initial concentration of the monomers was fixed at 0.1 M (equimolar) for the implementation of the fed-batch operation mode for the synthesis of POS. Then, by consecutive

stepwise addition of 0.1 M/h of each monomer, after 5 h, 0.5 M of each monomer was attained in the polymerization media. Simultaneously, the polycondensation in batch operation mode started with an initial monomer concentration of 0.5 M. Both polycondensation operation modes (batch vs. fed-batch) were evaluated in three different polymerization media: miniemulsion, water, and an organic solvent (cyclohexane:THF 5:1 *v/v*) (Table 4). The initial pH value of the miniemulsion and water was corrected to 5.0 before adding the biocatalyst (PBLI).

**Table 4.** The molecular weights of POS and conversion achieved in batch and fed-batch operation modes using PBLI (8 mg mL$^{-1}$) in the miniemulsion, water, and an organic solvent at 250 rpm and 45 °C.

| | Time (h) | Batch | | Fed-Batch | |
|---|---|---|---|---|---|
| | | $M_w$ (g mol$^{-1}$) | Conversion (%) | $M_w$ (g mol$^{-1}$) | Conversion (%) |
| Miniemulsion [a] | 1 | 2600 | 92.9 ± 1.0 | 2700 | 100 ± 0.5 |
| | 24 | 3700 | 98.2 ± 1.1 | 2700 | 76.9 ± 1.3 |
| | 48 | 7800 | 98.3 ± 0.7 | 4600 | 96.2 ± 1.0 |
| Water [a] | 1 | 2700 | 75.6 ± 1.8 | 1950 | 100 ± 0.4 |
| | 24 | 3450 | 98.5 ± 1.5 | 3450 | 100 ± 0.3 |
| | 48 | 6900 | 94.3 ± 1.3 | 3800 | 100 ± 0.5 |
| Organic solvent | 1 | 2700 | 53.0 ± 2.3 | 4500 | 74.4 ± 1.4 |
| | 24 | 5900 | 87.5 ± 1.9 | 4600 | 93.2 ± 0.9 |
| | 48 | 5900 | 86.7 ± 1.8 | 5900 | 94.6 ± 1.2 |

[a] With initial pH correction to 5.0.

High conversion and rapid polyester formation with molecular weights between 2600 and 2700 g mol$^{-1}$ occurred in the first hour of polycondensation for all polycondensation media in batch operation mode. After 24 h of polycondensation, the molecular weight in the miniemulsion and water was very similar at 3700 g mol$^{-1}$ and 3450 g mol$^{-1}$, respectively. The polymer molecular weight at 24 h was significantly higher in the organic solvent than the other two polymerization media (5900 g mol$^{-1}$). However, after 48 h, the polymer molecular weight in the organic solvent did not change from 24 h, while there was an increase of 52% and 50% in the miniemulsion and water, reaching 7800 and 6900 g mol$^{-1}$, respectively, after 48 h of polycondensation. Furthermore, a lower conversion (86.7%) was observed in the organic solvent (Table 4). These results clearly show that the biocatalyst reacts at different rates as a function of the monomer or polymer molecular weight already formed. In this case, PBLI showed a higher activity for monomers and short-chain polymer molecules. The enzyme activity slowed down due to the high sterical hindrance for longer-chain polymers accumulated in the three polycondensation media.

The highest conversion (98.3 ± 0.7%) and a molecular weight of 7800 g mol$^{-1}$ after 48 h of polycondensation were achieved in the miniemulsion in batch operation mode.

The POS synthesis in water at pH 5.0 showed a polyester molecular weight of 6900 g mol$^{-1}$, with high conversion (up to 94.3 ± 1.3%) after 48 h. A significant amount of monomers and polymer continued in solid precipitates due to their very low solubility in water but presenting a similar performance as that obtained in the miniemulsion. These results for the polycondensation of suberic acid and 1,8-octanediol at equimolar monomer concentrations (0.5 M) in water at an initial pH of 5.0 are superior to those obtained in the previous Section 3.2 with initial pH 3.3. These results are also superior to those obtained by several authors with the same monomer and 0.1 M substrate concentrations, with molecular weights and conversions of 1200 g mol$^{-1}$ and 12%, and 1600 g mol$^{-1}$ and 3%, respectively [32,56].

In comparison, the polycondensation in the organic solvent presented in this work the lowest values of polymer molecular weight and conversion after 48 h (5900 g mol$^{-1}$ and 86.7 ± 1.8%, respectively). For example, Azim 2006 [24] observed that the polyester precipitation in diphenyl ether during the polycondensation overcame increasing tempera-

tures from 80 °C to 95 °C, keeping the polyester soluble and proceeding with the enzymatic polyester synthesis in a monophasic polymerization medium.

The lower polycondensation efficiency in the organic solvent may be due to the low initial water content that negatively affects enzyme activity. Another possibility could be associated with the sequestration of water molecules from the enzyme microenvironment layer when using polar organic solvents like THF. The organic solvents are also usually associated with enzyme unfolding or relaxing of the right enzyme 3D structure and, consequently, lower enzyme activity [15].

The polymer molecular weights obtained in fed-batch operation mode were smaller than the ones assayed in batch operation mode for all polycondensation media after 48 h (Table 4). These results indicate that a high monomer concentration (0.5 M) may not have significant enzyme inhibition of the PBLI biocatalyst in POS synthesis in the miniemulsion and water but may negatively affect enzyme activity during polycondensation in the organic solvent (Table 4). This enzyme activity inhibition in the organic solvent is confirmed, as for the lowest monomer concentration (0.1 M) in the first hour of polycondensation in fed-batch operation mode, the polymer molecular weight reached a higher value (4500 g mol$^{-1}$) in comparison to batch operation mode (2700 g mol$^{-1}$) for the same time of polymerization.

Additionally, high POS solubility in the cyclohexane:THF (5:1 *v/v*) solvent could enhance the initial polymer molecular weight. The conversion difference achieved after 1 h of polycondensation of 74.4 ± 1.4% and 53 ± 2.3% for fed-batch and batch operation modes, respectively, also confirm the enzyme inhibition in the organic solvent for a high monomer concentration (0.5 M) (Table 4). Other authors have already observed a similar effect where the fed-batch operation mode protected the enzyme preparation against the inhibitory effect of a high substrate concentration in an organic solvent [47,55].

The fed-batch operation mode in the miniemulsion and water showed rapid and efficient formation of oligomers or small-chain polyesters after 1 h due to the high conversion (100 ± 0.5%) observed (Table 4).

Despite the better polycondensation performance of PBLI in fed-batch operation mode in the organic solvent after 1 h of reaction, equal molecular weights were achieved after 48 h of polycondensation (5900 g/mol$^{-1}$) in batch operation mode at 24 and 48 h. This result suggests that there is another factor affecting the PBLI activity negatively in the organic solvent.

These results showed that regarding the polymer molecular weight and conversion, the batch operation mode at equimolar monomer concentrations (0.5 M) is more appropriate for the synthesis of POS in the miniemulsion and water. In contrast, the fed-batch operation mode should be implemented for polycondensation in an organic solvent.

*3.4. Effect of Stirring Type on POS Synthesis*

Regardless of polymerization, the media must be homogeneously stirred for the full period of the polycondensation using an appropriate agitator design. Radial flow impellers are frequently used in emulsion polymerization at the laboratory scale [36].

The POS synthesis was tested in a small reactor containing an impeller (RI) constituted by a magnetic stirrer rod with a two-blade propeller and compared with the previously described glass vial reactor with a magnetic stirrer (RMS) to analyze the effect of stirring type on the polycondensation conditions. The polymerization working volume in the RI increases 5-fold (50 mL) compared to the RMS (10 mL). The polycondensation was performed in batch operation mode at equimolar monomer concentrations (0.5 M) and using PBLI (8 mg mL$^{-1}$) at 45 °C in the miniemulsion, water, and the organic solvent (cyclohexane:THF 5:1 *v/v*). The initial pH was corrected to 5.0 in the aqueous polycondensation media (miniemulsion and water) before adding the biocatalyst (Table 5).

**Table 5.** Samples withdrawn during the POS synthesis at 45 °C in the miniemulsion, water, and the organic solvent in terms of polymer molecular weight and conversion for direct magnetic stirring in a glass vial (10 mL) (reactor with magnetic stirring (RMS)) at 250 rpm and reactor with an impeller (50 mL) (RI) at 500 rpm.

| | Time (h) | Vial with a Magnetic Stirrer (RMS) | | Reactor with Impeller (RI) | |
|---|---|---|---|---|---|
| | | $M_w$ (g mol$^{-1}$) | Conversion (%) | $M_w$ (g mol$^{-1}$) | Conversion (%) |
| Miniemulsion [a] | 1 | 2690 | 92.3 ± 0.9 | 2690 | 45.3 ± 2.1 |
| | 2 | 2690 | 90.3 ± 1.0 | 2690 | 90.2 ± 1.4 |
| | 4 | 2690 | 93.4 ± 1.2 | 2690 | 95.0 ± 0.9 |
| | 8 | 2690 | 100 ± 0.6 | 2690 | 94.4 ± 0.9 |
| Water [a] | 1 | 2690 | 72.6 ± 1.4 | 2690 | 56.8 ± 2.0 |
| | 2 | 2690 | 98.1 ± 0.8 | 2690 | 91.5 ± 1.2 |
| | 4 | 2500 | 100 ± 0.4 | 2690 | 88.1 ± 1.0 |
| | 8 | 3460 | 100 ± 0.6 | 2690 | 99.3 ± 0.4 |
| Organic solvent | 1 | 2690 | 53.9 ± 2.0 | 2690 | 63.1 ± 1.8 |
| | 2 | 3460 | 72.5 ± 1.7 | 2690 | 77.7 ± 1.7 |
| | 4 | 4900 | 82.2 ± 1.1 | 2690 | 79.7 ± 1.0 |
| | 8 | 4900 | 86.7 ± 1.3 | 2690 | 78.8 ± 1.3 |

[a] With initial pH correction to 5.0.

The stirring rate of 250 rpm of the impeller was insufficient for efficient biocatalyst dispersion in the reactor with an impeller (RI), so the stirring rate of the impeller was enhanced to 500 rpm.

Despite the increase in the impeller stirring rate (500 rpm), the homogenization of polycondensation media after 1 h was inefficient inside the reactor with an impeller (RI) due to increased viscosity with polyester formation. Despite the higher solubility of the monomers and polyester in the organic solvent, the formation of a like-gummy was very viscous, which made the homogenization of the polycondensation media difficult inside the RI. Consequently, the enzyme–polymer–monomer contact was limited after 1 h of the POS synthesis in the RI. Additionally, stirring inside the RI during the POS synthesis started to be irregular between 1 and 8 h in the three polycondensation media. The impeller showed difficulty in moving and stopped several times, and for this reason, the polymerization was interrupted after 8 h. For comparison, the polycondensation in the RMS was also finished at 8 h.

The conversion after 1 h of polycondensation in the miniemulsion was higher (92 ± 0.9%) in the RMS in comparison to the RI (45 ± 2.1%). Yet, the same POS molecular weight was observed in both types of reactors (Table 5). The same tendency was observed for the polycondensation in water. In the organic solvent, a lower conversion for the RMS (53.9 ± 2.0%) compared with the RI (63.1 ± 1.8%) was observed. The polymer molecular weight stayed equal for both types of reactors.

The conversion in the miniemulsion (92.3 ± 0.9%) was higher than in water (72.6 ± 1.4%) and the organic solvent (53.9 ± 2.0%) in the first hour of polycondensation in the RMS, but in the RI, after 1 h, the conversion was lower (45.3 ± 2.1%) in the miniemulsion than in water (56.8 ± 2.0%) and the organic solvent (63.1 ± 1.8%) (Table 5). After 8 h of polycondensation in water, the polymer molecular weight was 3460 and 2690 g mol$^{-1}$ in the RMS and the RI, respectively (Table 5).

The conversion in the RI (78.8 ± 1.3%) after 8 h of polymerization in the organic solvent was lower, around 8%, compared to the conversion in the RMS and approximately 16% and 20% lower compared to the conversion in the RI for the miniemulsion and water, respectively. The polymer molecular weight obtained after 8 h in the organic solvent was 4900 g mol$^{-1}$ (RMS) and 2690 g mol$^{-1}$ (RI).

Even though the conversion increased slightly between 1 and 8 h for POS synthesis in the RI, the polymer molecular weight stabilized to about 2690 g mol$^{-1}$ for all polycondensation media tested (Table 5). These molecular weights are similar to those achieved in Section 3.3 with direct magnetic stirring (250 rpm) in glass vials (RMS) in the miniemulsion

and water (2600 g mol$^{-1}$ and 2700 g mol$^{-1}$, respectively) after 1 h polycondensation. However, they are inferior than the POS molecular weights obtained in these polymerization media after 48 h (5900 and 7800 g mol$^{-1}$, respectively) (Table 4). These results suggest poor performance of the polycondensation reaction in the RI for 8 h due to high viscosity and deficient homogenization of the biocatalyst independent of the polycondensation media tested.

The geometry, diameter reactor/impeller ratio, type of impeller, and viscosity of the polycondensation media should be optimized due to the complexity of these parameters and their effect on mass transfer resistance, mainly when using a biocatalyst constituted by an enzyme immobilized on a support surface. Meyer (2003) [57] confirmed that "scale-up procedures are not trivial, and special attention has to be paid to this viscosity increase during polymerization, decreasing the heat- and mass-transfer rates" [58]. Therefore, different reactor/impeller geometry and type of impeller (e.g., marine propeller) are fundamental to be studied and optimized, considering these performance limitation factors on POS synthesis in this work.

### 3.5. Reutilization of the PBLI Biocatalyst during POS Synthesis in the Reactor with an Impeller (RI)

Despite the lower performance of the POS synthesis observed in the reactor with an impeller (RI), the RI was chosen to evaluate the reutilization of the PBLI biocatalyst as the stainless steel network can retain the biocatalyst totally at the end of each reutilization cycle. This way, the biocatalyst loss was avoided if the filtration was performed outside the reactor, and it could contribute to a significant experimental error, especially when working with a small amount of biocatalyst.

Based on the assumption that an increase in polymerization temperature decreases the viscosity, the polycondensation temperature was tested now at 60 °C in the small reactor with an impeller (RI) for up to 2 h of POS synthesis since no significant variation of the polymer molecular weight and conversion was observed for a more extended period (Table 5).

Additionally, the PBLI reutilization (8 mg mL$^{-1}$) through successive cycles of the POS synthesis was tested. The reuse of this biocatalyst was performed and compared in the miniemulsion, water, and the organic solvent system (cyclohexane:THF 5:1 *v/v*) in batch operation mode at an equimolar substrate concentration (0.5 M) at 500 rpm and 60 °C and initial pH 5.0 in the miniemulsion and water (Figure 6). The immobilized enzyme preparation (PBLI) was reused for seven successive cycles. The polycondensation time for the three media was 2 h in the first five cycles and 24 h and 48 h for the sixth and seventh cycles, respectively. This way, the immobilized enzyme preparation (PBLI) operational stability was evaluated for a more extended period at 60 °C. The efficiency of enzyme reutilization was expressed through the dicarboxylic acid conversion for the three polymerization media (Figure 6).

Despite the high viscosity in all polycondensation media, the impeller inside the small reactors showed adequate homogenization under these polymerization conditions (60 °C and 500 rpm).

Figure 6 shows that PBLI could be reused efficiently in the miniemulsion and water at least seven cycles corresponding to 72 h at 60 °C without significant polymer molecular weight variation (around 2700 g mol$^{-1}$) and conversion higher than 90% (Figure 6). The miniemulsion comprised 81% of water, promoted high enzyme stability, and minimized the monomers' enzyme inhibitory effect observed in the organic solvent. These results also showed that the enzyme was not desorbed from the support surface inclusively at 60 °C despite several incubation cycles of this biocatalyst with Triton X-100 as a surfactant used in the miniemulsion formulation.

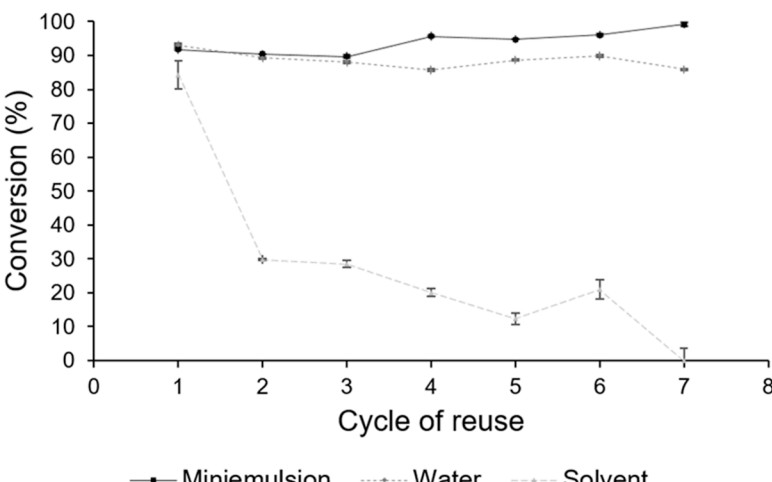

**Figure 6.** Conversion in successive cycles of POS synthesis at 60 °C using PBLI (8 mg mL$^{-1}$) for 7 cycles (equivalent to 72 h) in the small reactor (70 mL) in batch operation mode in the miniemulsion, water, and the organic solvent at 500 rpm.

The biocatalyst stability in water was also high, and as the monomers are in solid state (precipitated) during the time course of polycondensation, the inhibitory effect was also insignificant (Figure 6).

In contrast, the loss of the catalytic capacity of the immobilized enzyme preparation (PBLI) in the organic solvent (cyclohexane:THF 5:1 *v/v*) was evident. After the second cycle of PBLI reuse in this organic solvent, only about 35 ± 0.2% of conversion was observed after polycondensation for 2 h. A total lack of conversion and no polymer formation occurred after the sixth cycle (Figure 6). Other authors have confirmed this lack of stability of biocatalysts in an organic solvent when reused in consecutive batch cycles of reaction. For example, Lerin (2011) [59], using Novozym 435, showed lower conversion (around 10%) after the sixth cycle for 2-ethylhexyl palmitate synthesis and 50% after the eighth cycle of reuse for ascorbyl palmitate synthesis in *tert*-butanol. For 1-glyceryl benzoate production using 2-propanol as the organic solvent, 20% conversion was observed after the tenth cycle of reuse [57].

The low capacity of reutilization of the PBLI biocatalyst during POS synthesis in the organic solvent can be due to the enzyme desorption from the support surface in contact with the mixture of cyclohexane and THF (5:1 *v/v*) at 60 °C as this lipase is likely physisorbed onto the support mainly through hydrophobic interactions [59]. However, this solvent is primarily composed of cyclohexane with a high log *P* (~3.4), and during filtration, a precipitate that could reveal the PBL desorption was not observed. Additionally, the $^1$H NMR spectrum (Figure 7) did not reveal any peak associated with protein trace contaminating the synthesized polyester in the organic solvent (cyclohexane:THF 5:1 *v/v*) (Section 3.6.1).

The low water content in the cyclohexane:THF (5:1 *v/v*) solvent, high temperature (60 °C), and high monomer concentration (0.5 M) may decrease enzyme activity and stability after the second cycle of polycondensation in the organic solvent.

*3.6. Polymer Characterization*

3.6.1. $^1$H NMR Analysis of the Polymer Molecular Structure (POS)

The POS samples produced in the miniemulsion, water, and the organic solvent were assayed by $^1$H NMR analysis to prove the polymer molecular structure and polymer purity (Figure 7A). NMR spectra peaks relative to the ester hydrogen bond in POS agree with the literature, showing a small displacement due to the different sample concentrations [60].

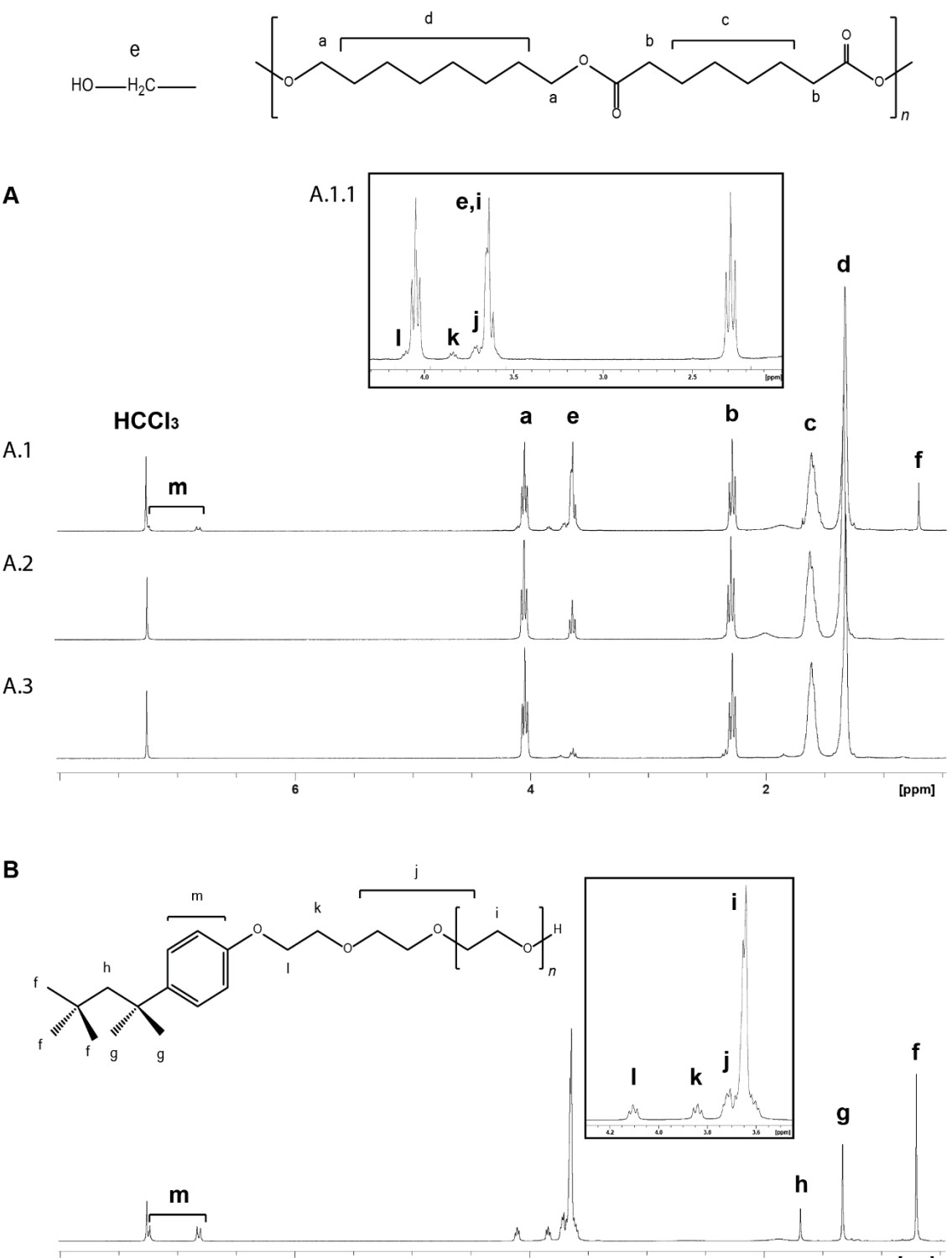

**Figure 7.** Nuclear magnetic resonance ($^1$H NMR) spectrum of POS samples obtained in (**A**) miniemulsion (A.1), water (A.2), and the organic solvent (A.3). The spectrum (**B**) represents the pure Triton X-100 surfactant.

In the $^1$H NMR spectrum of poly(octylmethylene suberate), it was possible to identify a diol (**d**) methylene multiplet at 1.30–1.45 ppm, a diacid (**c**) methylene multiplet at 1.55–1.70 ppm, and a methylene multiplet adjacent to the carbonyl group (**b**) at 2.30–2.40 ppm (Figure 7A). Additionally, a methylene triplet adjacent to the hydroxide group (**e**) at 3.60–3.70 ppm corresponding to the C-linked Hs of the terminal hydroxyl of the polymer may

also refer to the unreacted dialcohol or dialkyl terminal hydroxyl and also a methylene triplet adjacent to the oxygen of the ester group (**a**) at 4.00–4.20 ppm (Figure 7A).

These peaks, **a**, **b**, **c**, **d**, and **e**, are intimately related to the molecular structure of POS (Figure 7A). In contrast, peaks associated with dicarboxylic acid are not observed due to the high conversion in all polymerization media and low solubility of this monomer in the analytical solvent (deuterated chloroform).

However, the polyester spectrum obtained in the miniemulsion (Figure 7A.1) showed additional NMR peaks (**f**, **i**, **j**, **k**, **l**, and **m**) compared to the spectrum obtained in water and the organic solvent (Figure 7A.2, Figure 7A.3, respectively). The additional peaks indicated the presence of Triton X-100, the surfactant used in the miniemulsion preparation, when compared to the $^1$H NMR spectrum of pure Triton X-100 (Figure 7B). This surfactant was not removed from the synthesized polyester by simple washing with water. The peak (**m**) at 6.8 ppm present in the miniemulsion spectrum is associated with the aromatic group of Triton X-100. As there is not a peak in the region between 6 and 10 ppm in the spectrum obtained in water and the organic solvent (Figure 7A.2, Figure 7A.3, respectively), and it is a good indication of the absence of an aromatic amino acid signal, there was no protein desorption from the PBLI biocatalyst support.

### 3.6.2. Thermogravimetric Analysis (TGA) of the POS Synthetized

The thermogravimetric analysis of the POS synthesized allows the verification and suggestion of this polymer's potential applicability [61]. The thermograms and the data of maximum temperature ($T_{max}$), the temperature of the extrapolated beginning ($T_{onset}$), and the melting temperature ($T_m$) of the POS synthesized in the miniemulsion, water, and the organic solvent are shown and presented in Figure 8 and Table 6.

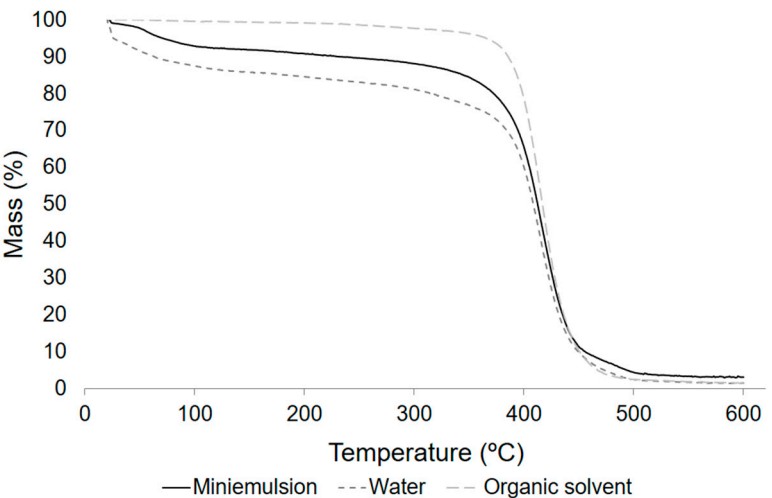

**Figure 8.** Thermograms of POS samples withdrawn from the miniemulsion, water, and organic solvent after 48 h of polymerization.

**Table 6.** Thermogravimetric analysis (TGA) of POS samples withdrawn from different polymerization reaction media in batch operation mode.

| System | $T_{max}$ (°C) | $T_{onset}$ (°C) | $T_m$ (°C) [2] | Weight Loss (%) |
|---|---|---|---|---|
| Miniemulsion | 488.6 | 387.7 | 59.2 | 97.5 |
| Water | 419.9 | 385.6 | 63.9 | 98.2 |
| Organic solvent [1] | 416.8 | 385.8 | 70.8 | 99.0 |

[1] Mixture of cyclohexane and tetrahydrofuran (THF) (5:1 *v/v*). [2] Obtained from the TGA heating curve.

The thermograms of the POS obtained in the miniemulsion and water show an initial weight loss at temperatures between 20 and 120 °C that are related to the evaporation of

traces of water captured inside the POS polymer chain during the enzymatic polymerization and not eliminated during the drying process (Figure 8).

The thermogram of the POS synthesized in the organic solvent did not show initial degradation probably due to the absence of cyclohexane, THF, and water (Figure 8).

The values of $T_{onset}$ (~386 °C) are very similar for the POS synthesized in the miniemulsion, water, and the organic solvent, which suggests that the thermal stability of POS is not dependent on the polycondensation media used (Table 6).

Furthermore, the surfactant contaminating the POS synthesized in the miniemulsion is responsible for the second degradation temperature of around 450 °C detected in the TGA thermogram (Figure 8). Additionally, the trace of Triton X-100 in the polyester can explain the higher $T_{max}$ and lower $T_m$ of the POS synthesized in the miniemulsion compared to polymerization in water and organic solvent systems (Table 6). TGA confirms the detection of Triton X-100 by [1]H NMR analysis (Figure 7) that may be an impurity in the polyester synthesized in the miniemulsion and may not be eliminated by a simple washing process at the end of the polycondensation reaction.

### 3.6.3. Differential Scanning Calorimetry (DSC) of the POS Synthetized

DSC analysis was performed to confirm the melting temperature and the ease of processing the poly(octylmethylene suberate) synthesized in the different polymerization media. The thermal properties of the polyesters are shown in Figure 9.

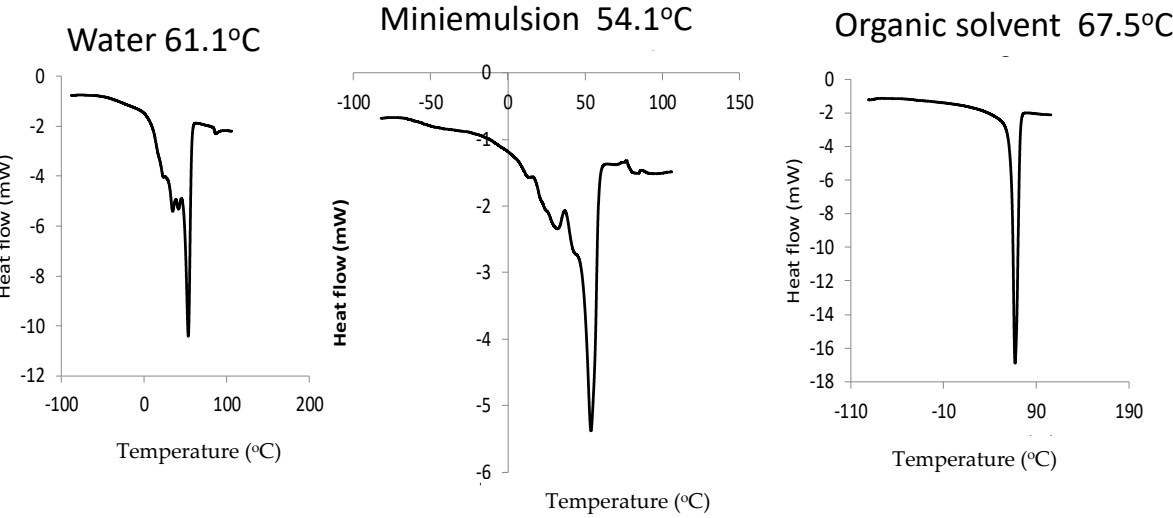

**Figure 9.** Thermal properties obtained by differential scanning calorimetry (DSC) of POS synthesized in the miniemulsion, water, and the organic solvent, presenting peak melting temperatures, respectively, of 54.1, 61.1, and 67.5 °C.

These results confirmed the melting points obtained by thermogravimetric analysis (TGA). The $T_m$ of the POS synthesized in the miniemulsion was lower than that obtained with polymerization in water and the organic solvent due to the Triton X-100 contamination, and it is in agreement with [1]H NMR and TGA results. Triton X-100 interferes with the crystalline lattice of the polyester, making it easier to break down the intramolecular forces between molecules, like in a pure crystalline polymer characterized by a very well defined melting point [59]. Consequently, less heat is required to break down these intermolecular interactions, and the melting point of the POS synthesized in the miniemulsion is lower than that of the POS produced in water and the organic solvent.

The polyesters synthesized in this work in different polymerization media proved to be crystalline since the glass transition temperature ($Tg$) was not observed for these polyesters under reported conditions. In turn, the melting temperatures were much lower

than those of commercially available polyesters, except for poly-$\varepsilon$-caprolactone (PCL), probably due to the crystalline organization of the synthesized polyesters.

## 4. Conclusions

The use of an immobilized enzyme preparation (PBLI) in the miniemulsion and water (pH of 5.0) proves to be an efficient method for poly(octamethylene suberate) (POS) synthesis with high polymer molecular weight and conversion. Furthermore, the POS synthesis in water and the miniemulsion showed excellent enzymatic stability of the PBLI biocatalyst, confirmed by the high and stable conversion obtained during the seven cycles of reuse (equivalent to 72 h at 60 °C). Additionally, water as a green solvent could be advantageous for industrial processes in polycondensation.

The polycondensation reaction performed in the organic solvent, despite a high log $P$ (~3.4) of cyclohexane, interfered negatively with the PBLI activity and showed a high negative impact on polyester synthesis and biocatalyst reutilization.

The batch operation mode is the better choice for POS synthesis in miniemulsions and water, and the fed-batch operation mode is better for polycondensation in an organic solvent.

The stirring type during the POS synthesis is shown to be an essential parameter to be carefully studied and optimized for large-scale reactors for polycondensation in miniemulsions and water.

According to this experimental work, the optimization of the geometry of the reactor for different impellers is fundamental to obtain some guidelines for scale-up and, in future work, enhance the polymer molecular weight. Polyesters with a molecular weight higher than 10,000 g mol$^{-1}$ are an important class of polymers widely used in producing fibers, films, and 3D structures. The other possibility is developing a two-step polymerization method combining direct polycondensation in aqueous media described in this work with low-temperature post-enzymatic polymerization of the pre-polymers previously obtained with less and well-controlled content of water.

## 5. Patents

The results reported in this manuscript are partially included in the patent approved in Portugal, PT 116045—Sintese de poliésteres em sistemas de polimerização em meio aquoso "de sólido para sólido" via biocatálise, de 31 December 2019, and constitutes one part of the new greener polymer-5B technology core.

English title—PT 116045—Synthesis of polyesters in aqueous polymerization media "from de solid to solid" via biocatalysis, 31 December 2019.

**Author Contributions:** Conceptualization, analytical work, data handling, writing original draft, and writing—review and editing: A.C.D.P.; conceptualization, writing—review and editing, and supervision: D.P.C.d.B.; conceptualization, methodology, data handling, writing—review and editing, resources, funding acquisition, and supervision L.P.F. All authors have read and agreed to the published version of the manuscript.

**Funding:** Funding was received by Ana C. D. Pfluck from the Coordenação de Aperfeiçoamento de Pessoal de Nível Superior (CAPES), Brazil, and the Institute for Bioengineering and Biosciences (iBB) from the Portuguese Foundation for Science and Technology (FCT; UID/BIO/04565/2013) and from the Programa Operacional Regional de Lisboa 2020 (project no. 007317).

**Institutional Review Board Statement:** Not applicable.

**Informed Consent Statement:** Not applicable.

**Data Availability Statement:** Not applicable.

**Conflicts of Interest:** The authors declare no conflict of interest.

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
