# Peer review of "Biodegradable Polyester Synthesis in Renewed Aqueous Polycondensation Media: The Core of the New Greener Polymer-5B Technology"

_processes, doi:10.3390/pr9020365_

Round 1
Reviewer 1 Report
The manuscript entitled "Biodegradable polyester synthesis in renewed aqueous polycondensation media: the core of the new "greener" Polymer-5B technology", and numbered processes-1081235, reports very intresting data.
However, this work could be published only after a major revision; therefore, authors should improve their manuscript as follows:
(a) The text is not easy to be followed due to a lot of syntax errors, as well as many incomprehensible sentences (within the text). A throrough reconsideration, improving and rewriting of the text is strongly recommended, by consulting an English spoken person.
(b) The Introduction is very long. It seems to me that authors just chatter. Therefore, Introduction should be shortened in order to offer the necessary introductory information to the reader, according to the methods which are followed in the text, the results and the conclusions.
(c) All methods should be rewritten carefully by describing clearly and accurately their steps (one by one).
(d) Although authors refer in the Abstract, as well as in the main text that they perform some kind of optimization of several experimental parameters, however what the reader receives is something incomprehensible, which has nothing to do with a parameter optimization process, which is based on robust statistical methodology. There are within the relative literature many examples, and authors are reccomended and encouranged to consult them and proceed accordingly, in order to improve the quality of their manuscript[1].
Overall, I recommend the publication of this manuscript after a major revision, and reconsideration, by providing an adequate time of about one month to the authors.
[11] (Examples) Current Biotechnology, 2013, 2, 23-29; Bioresource Technology 238 (2017) 122–128, etc
Reviewer 2 Report
The paper reported by Pfluck et al. deals with an investigation on the lipase-catalyzed-polycondensation between aliphatic carboxylic diacids and dialcohols using free and immobilized enzyme preparations. Optimization of reaction conditions was carried out in order to recognize the best condensation system for the synthesis of biodegradable polyesters.
The paper presents many interesting data and I feel is fine for publication on Processes.
However, I suggest reviewing English all over the text and making attention to presentation of pictures (e.g. figure 1 is not complete) and graphics.
I would also like to point out a brief opinion on the introduction section: in my candid opinion introduction is too long and honestly is hard to have a clear idea of the logical plan that authors have in mind. This reviewer believes that a good plan needs to be explained in few words, while long and extensive paragraphs often mask a weakness in the paper structure. However, I believe this is not the scope of the authors, so please reduce the content and makes it more clear in fewer words. Maybe a schematic representation of the plan could be useful to give an immediate positive feedback to the readers, which would be more encouraged to read the remaining part of the paper.
In conclusion, a double check of data presentation and english is required for the manuscript before consideration foo publication on Processes.
Round 2
Reviewer 1 Report
The revised version of the manuscript entitled: Biodegradable polyester synthesis in renewed aqueous polycondensation media: the core of the new "greener" Polymer-5B technology, and numbered processes-1081235, can be published in the journal Processes